# Estimation of Aboveground Biomass Stock in Tropical Savannas Using Photogrammetric Imaging

**Roberta Franco Pereira de Queiroz [1,*], Marcus Vinicio Neves d'Oliveira [2], Alba Valéria Rezende [1] and Paola Aires Lócio de Alencar [1]**

[1] Department of Forestry, Campus Darcy Ribeiro, University of Brasília, Brasilia 70910-900, DF, Brazil; albavr@unb.br (A.V.R.); paolaaires12@gmail.com (P.A.L.d.A.)

[2] Brazilian Agricultural Research Company (EMBRAPA), Agroforestry Research Center of Acre, Rio Branco 69900-180, AC, Brazil; marcus.oliveira@embrapa.br

[*] Correspondence: roberta.queiroz@icmbio.gov.br

**Abstract:** The use of photogrammetry technology for aboveground biomass (AGB) stock estimation in tropical savannas is a challenging task and is still at a preliminary stage. This work aimed to use metrics derived from point clouds, constructed using photogrammetric imaging obtained by an RGB camera on board a remotely piloted aircraft (RPA), to generate a model for estimating AGB stock for the shrubby-woody stratum in savanna areas of Central Brazil (Cerrado). AGB stock was estimated using forest inventory data and an allometric equation. The photogrammetric digital terrain model (DTM) was validated with altimetric field data, demonstrating that the passive sensor can identify topographic variations in sites with discontinuous canopies. The inventory estimated an average AGB of 18.3 ($\pm$13.3) Mg ha$^{-1}$ at the three sampled sites. The AGB model selected was composed of metrics used for height at the 10th and 95th percentile, with an adjusted R$^2$ of 93% and a relative root mean squared error (RMSE) of 16%. AGB distribution maps were generated from the spatialization of the metrics selected for the model, optimizing the visualization and our understanding of the spatial distribution of forest AGB. The study represents a step forward in mapping biomass and carbon stocks in tropical savannas using low-cost remote sensing platforms.

**Keywords:** remote sensing; Cerrado; vegetation; mapping; carbon stock; forest monitoring

## 1. Introduction

Savannas are characterized by the coexistence of a continuous stratum covered by grasses and a discontinuous tree stratum within a broad environmental gradient [1]. Savannas occur in regions with strong seasonality; their form of vegetation typically occurs in transition regions between humid and semi-arid environments [2] and is influenced by the availability of water and soil resources, the frequency of fires, and the presence of large grazing herbivores in their evolutionary history [3].

The Brazilian savanna, known as Cerrado, is the second largest biome in Brazil ($\cong$2 million km$^2$) and represents the largest neotropical savanna formation in America [1]. It is considered the most biodiverse savanna in the world [4]. The Cerrado vegetation presents several physiognomies, varying between grass, savanna, and forests [5], although the dominant vegetation of the biome is a wooded savanna termed cerrado sensu stricto. As the Cerrado occupies a large area of the Brazilian national territory, it plays an important role in mitigating climate change, storing carbon in its biomass and in the soil. However, the Cerrado has one of the highest national rates of deforestation and degradation, due to the expansion of agricultural and infrastructure activities [6,7].

Aboveground biomass (AGB) is an essential variable for quantifying carbon stock and its sequestration from the atmosphere, offering a strategic ecosystem service for political decision-making [8]. Accurate AGB estimates depend mainly on efficient methods for obtaining data and monitoring over time [9]. Nevertheless, monitoring and estimates of

AGB in the Cerrado present a complex and challenging task, due to the great diversity of its physiognomy and species [10]. In this context, expanding the application of new technologies and methodologies for estimating AGB, which can be both accurate and economically viable, is challenging for Cerrado savanna formations.

In recent decades, the combination of remote sensing tools and imaging technologies has allowed the three-dimensional reconstruction of vegetation and has enabled the mapping of the spatial variability of forests, estimating their AGB stocks and structure, and contributing to biodiversity monitoring [11–13]. However, in a fragmented landscape, where the average size of the study areas is small, high-resolution images are necessary [14]. For these smaller scales, spaceborne satellite remote sensing alone does not meet the requirements, due to the costs of acquiring images of high spatial resolution, and also because of possible negative impacts as a result of weather conditions [13,14].

In this context, remotely piloted aircraft (RPA) represent the most suitable type of platform for application on smaller scales with local and regional objectives. RPAs offer high cost-effectiveness when compared to the use of satellite imagery for small project applications [15,16]. Moreover, RPAs offer flexibility in their flight plan, can generate data on demand (avoiding unfavorable weather), can be used for real-time operations (such as fire monitoring), and can optimize both time and labor [11,12,14]. Passive sensors (RGB cameras) have become viable and lower-cost alternatives for small-scale forest assessments [17,18] in remote regions and for developing countries [16]. Passive sensors on board RPAs can capture digital images with high overlap; from those images, photogrammetric algorithms generate digital models, enabling a range of environmental applications [19–21].

An accurate digital terrain model (DTM) is essential for these forest analyses, as it makes it possible to normalize vegetation height and produce accurate metrics related to the vertical structure of the forest [22,23]. Passive sensors have the limitation of being unable to penetrate the canopy and identify the ground surface [15]. This limitation is even greater in tropical forests with dense canopies and in areas with high topographic variation [24]. Nevertheless, in areas with sparse and open canopy cover, such as savannas, passive sensors can produce a reliable DTM [16,22].

Savanna ecosystems are challenging from the remote sensing perspective, due to their inherent mix of discontinuous woody cover and herbaceous vegetation [13]. Especially in the case of the Cerrado, the use of new remote sensing technologies in the estimation and monitoring of carbon and biomass stocks is still quite scarce [25,26]. Although new studies have been developed in recent years, along with the application of photogrammetry techniques using passive sensors in savannas [27], from the perspective of the Cerrado, these kinds of studies are at a preliminary stage [28,29].

Our objective was to generate an AGB estimation model for the shrubby-woody stratum of cerrado sensu stricto physiognomy, using a RGB camera on board an RPA. Considering that the potential for generalization of some estimation models is limited, due to approaches that use site-specific data [14], in this study, we selected three different sites with different land-use histories. We thus incorporated a broad representative range of vegetation into the data collection process and addressed two key questions: (1) Can a passive sensor combined with photogrammetry describe the topographic variation through the digital terrain model? (2) Is it possible to accurately estimate and map the AGB stock of Cerrado savanna vegetation and, thus, enable the use of these low-cost technologies to optimize vegetation monitoring and management?

To this end, using forest inventory data collected on ground plots in cerrado sensu stricto sites and images of these plots obtained by an RGB camera on board an RPA, we: (i) produced and validated a digital terrain model generated by photogrammetry using topographic field data; (ii) used the generated point cloud metrics to produce a linear regression model for estimating biomass stock; (iii) mapped the AGB distribution in the savannas areas of the Cerrado that were flown over by the RPA and then studied.

## 2. Materials and Methods

### 2.1. Study Areas

The study was conducted in three Cerrado sites located in Brazil's Federal District, comprising two sites within the Águas Emendadas Ecological Station (ESECAE) and one site within the Água Limpa experimental farm (FAL) (Figure 1). All areas are classified as cerrado sensu stricto, a woodland savanna formation characterized by defined tree and shrubby-herbaceous strata, with trees distributed randomly and in different densities [5]. Cerrado sensu stricto also represents 70% of the Cerrado biome. This predominant landscape physiognomy occurs on extensive high plateaus and is a type of physiognomy that has lost most of its areas due to land use change [30]. The regional climate is that of typical savanna (Aw, according to Köppen), with an average annual temperature of 21 °C and average annual precipitation of 1500 mm [31].

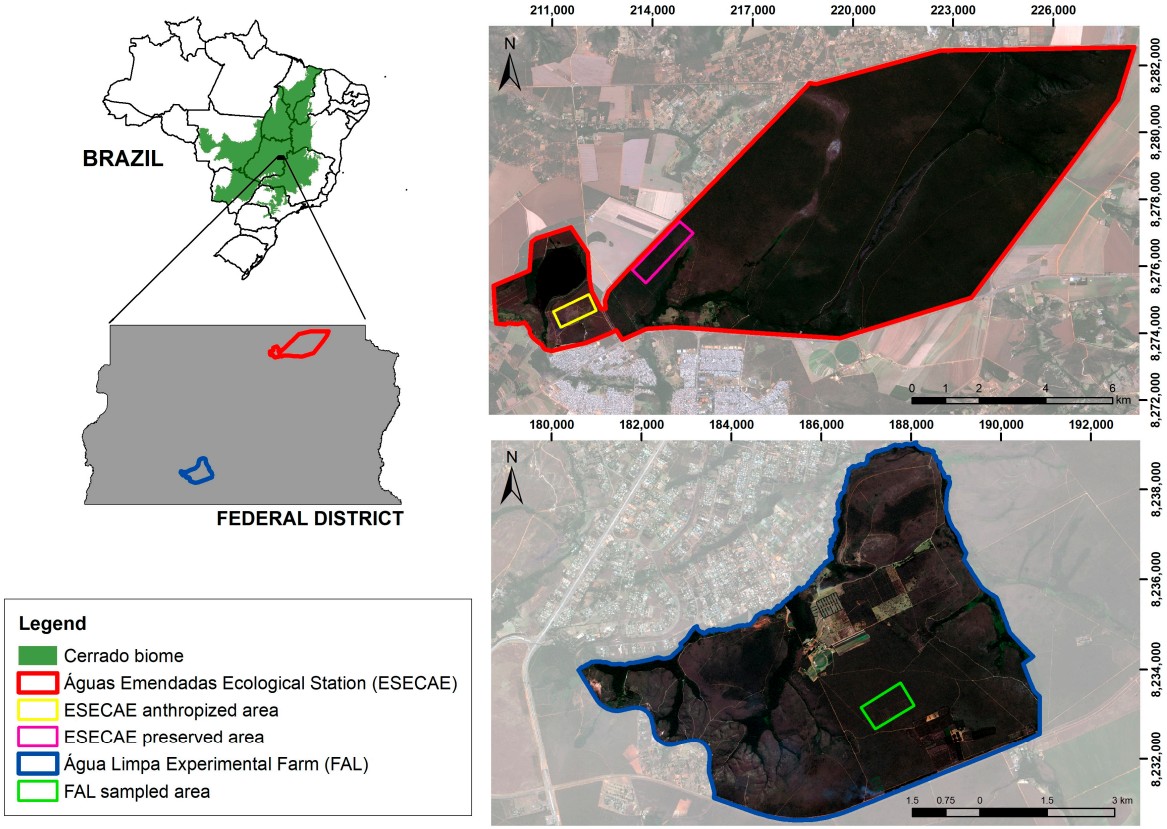

**Figure 1.** Location of the study areas of cerrado sensu stricto in the Federal District, Brazil.

ESECAE is a fully protected Conservation Unit (CU) located in the extreme northeast of the Federal District, with a total area of 10,547 ha, and is located between the coordinates 15°31′–15°35′ S and 47°31′ and 47°42′ W. It is situated at an average altitude of 1100 m and the main soil classes occurring in the area are red latosols and red-yellow latosols [32]. Two Cerrado sites in different stages of succession were sampled within the ESECAE area.

The first site, which has been intensely anthropized, has been undergoing natural regeneration since 1988 (Figure 1). Originally occupied by the cerrado sensu stricto, the area was then used for cattle ranching and agriculture. Eighteen years after the regeneration process began, the area has recovered only 15% of its woody species and there is a large presence of competing invasive grasses [33]. The second site is a preserved area and represents intact cerrado sensu stricto without disturbance (Figure 1).

FAL occupies about 4340 ha and lies at an average altitude of 1100 m. The site is located between the geographic coordinates 15°56′–15°59′ S and 47°53′–47°59′ W, at a mean elevation of 1159 m (Figure 1). This is an experimental area in which six silvicultural

treatments were implemented in 1988 in three blocks, i.e., with three repetitions. Each experimental plot was surrounded by Cerrado vegetation; through constant monitoring, it was found that all the experimental plots were able to recover their volume and above-ground carbon stocks after 23 years of regeneration, the final levels being similar to the stocks of the area prior to silvicultural treatments [34].

The sites sampled, although representing the same cerrado sensu stricto physiognomy, differ in their use, fire history, and stage of succession. Therefore, they present great variability in terms of configuration in the AGB of the shrubby-woody stratum (Figure 2), characterizing a wide range of representativeness of Cerrado areas in the Federal District.

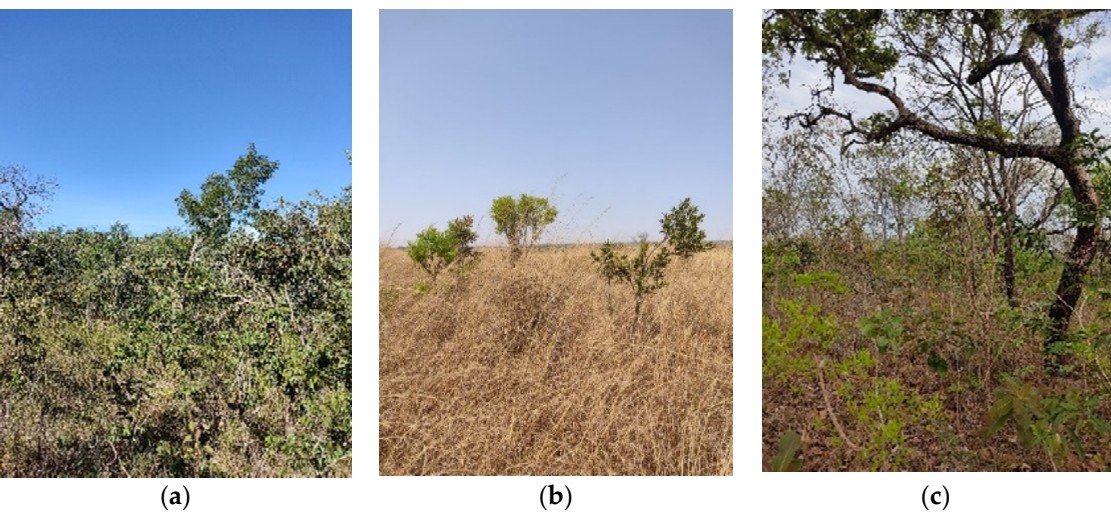

(**a**)                  (**b**)                  (**c**)

**Figure 2.** Field view of the cerrado sensu stricto sites sampled in the Federal District, Brazil, namely: (**a**) experimental area in the Água Limpa experimental farm (FAL); (**b**) anthropized area in ESECAE; (**c**) preserved area in the ESECAE unit.

### 2.2. Forest Inventory

A forest inventory was conducted using randomly distributed plots (20 m × 50 m), with ten and nine plots sampled in the anthropized and preserved sites of ESECAE, respectively, and eighteen plots in FAL. Considering all three areas, 37 plots were measured, totaling 3.7 ha of sampled area. The forest inventory was conducted at FAL in September 2019 and at ESECAE in October 2021 (Table 1). The diameter and height of all woody individuals (both living and dead (standing)) of the tree and shrub stratum (shrubby-woody stratum) with a diameter at base height (Db, measured at 30 cm above ground) that was greater than or equal to 5 cm were measured. All individuals were botanically identified at the genus, species, and family levels.

**Table 1.** Summarized data collected from the forest inventory at three cerrado sensu stricto sites in the Federal District, Brazil.

|  | FAL | ESECAE Anthropized Area | ESECAE Preserved Area |
|---|---|---|---|
| Mean diameter (cm) | 7.7 ± 2.8 | 8.1 ± 3.6 | 10.5 ± 5.8 |
| Mean height (m) | 3.1 ± 1.2 | 3.3 ± 1.2 | 4.1 ± 1.8 |
| Total species richness | 74 | 19 | 67 |

### 2.3. Aboveground Estimates (AGB)

Estimates of AGB were obtained by summing the estimates of each individual recorded tree within each plot. AGB was estimated using Equation (1), an allometric equation developed and adjusted for the cerrado sensu stricto of the Federal District [10]:

$$AGB = 0.49129 + 0.02912 \times Db^2 \times Ht \tag{1}$$

$$R^2 \text{ (\%)} = 98.28; \text{ RMSE (\%)} = 25.79$$

where B is dry biomass (kg); Db is the diameter measured at 0.30 m from ground level (cm) and Ht is the total height (m); $R^2$ (%) is the coefficient of determination as a percentage and RMSE (%) is the root mean squared error as a percentage.

### 2.4. Overflight with Remotely Piloted Aircraft (RPA)

The RPA used in this study was a DJI Phantom 4 RTK multi-rotor model, coupled with an RTK module, equipped with a 1″ CMOS (complementary metal-oxide semiconductor) multispectral camera of 20 megapixels per square inch and a 24-millimeter autofocus lens. The D-RTK2 mobile station is equipped with global navigation satellite system (GNSS) receivers and provides real-time positioning data for each image captured, ensuring centimeter precision for the images and results generated. At FAL, the RPA overflight took place in October 2020, while the flight at ESECAE took place in October 2021.

The flights were performed autonomously, at a speed of 45 km/h, as a nadir flight (camera pointing downwards), with 85% frontal and lateral overlap. The aerial images were processed in Pix4DMapper software [35], by means of the feature detection algorithm SIFT (scale-invariant feature transform) [36] and were processed using structure from motion (SfM) methods to generate the point clouds and produce three-dimensional (3D) information [11,15]. The processing resulted in the production of the digital surface models (DSM) and digital terrain models (DTM).

At each plot vertex, white targets were embedded that were made of cardboard and sulfite paper. The maximum reflectance of the targets' white color served to better visualize the vertices of the plots and their locations on the aerial images. The spatial resolution of the image is influenced, among other factors, by the focal length of the camera and the height of flight. This resolution is defined as ground sample distance (GSD), which indicates how much area an image pixel represents in terms of centimeters on the ground [15,37]. Overflights were performed at a height of 90 m, resulting in an average GSD of 5 cm.

### 2.5. Digital Terrain Model (DTM) Validation

Ground control points (GCPs) were collected in each sample plot vertex. The GCPs capture geolocation information (X, Y, and Z), which allowed us to verify the correlation of the altimetric variable Z from the GCPs with the altitude information from the digital terrain model (DTM) generated via photogrammetry. A total of 77 GCP points were collected at the FAL site, while 54 were collected at the ESECAE unit, with 25 points in the preserved site and 29 in the anthropized site.

TechGeo Zênite II equipment was used, along with global navigation satellite system (GNSS) receivers of dual frequency (L1 and L2) and dual constellation (GPS and GLONASS). Each point had an acquisition time of 10 min, following a method adapted from the work of d'Oliveira et al. [38]. The receiver data were post-processed using GTR processor software [39], based on data from the Brasilia base station (BRAZ91200), which is the reference station of the Brazilian Continuous Monitoring Network (RBMC).

Using ArcMap software [40], the corresponding altimetric values in each GCP were extracted from the DTMs. These values were tested for normality, correlation, and statistical difference.

### 2.6. Adjustment and Validation of the Mathematical Model

The digital terrain model was then subtracted from the digital surface model to remove topographic variation within the plot, resulting in a canopy height model (CHM) that presents metrics of the vertical structure of the forest (Table 2). The metrics of the CHM were extracted at the plot level using Fusion software [41], with one meter being established as the minimum height to exclude those values referring to the grassy layer present in the cerrado sensu stricto environment.

**Table 2.** Metrics taken from the digital images captured with remotely piloted aircraft, obtained with photogrammetry techniques, and used for modeling aboveground tree biomass in the Cerrado areas of the Federal District.

Maximum Height
Average Height
Height Mode
Median heights
Height Standard Deviation
Height Variance
Height variation coefficient
Mean heights from 1st to 4th quartile
Coefficient of variation, kurtosis, and skewness of the mean heights of the quartiles
Mean heights of the 1st, 5th, 10th, 20th, 25th, 30th, 40th, 50th, 60th, 70th, 75th, 80th, 90th, 95th, and 99th percentiles
Quadratic mean height
Mean cubic height
Canopy cover ratio

Plot-level CHM metrics were merged with the summarized field plot data for regression modeling (Figure 3). Multiple linear regression techniques were used to develop the relationships between plot-level CHM metrics and field-measured AGB.

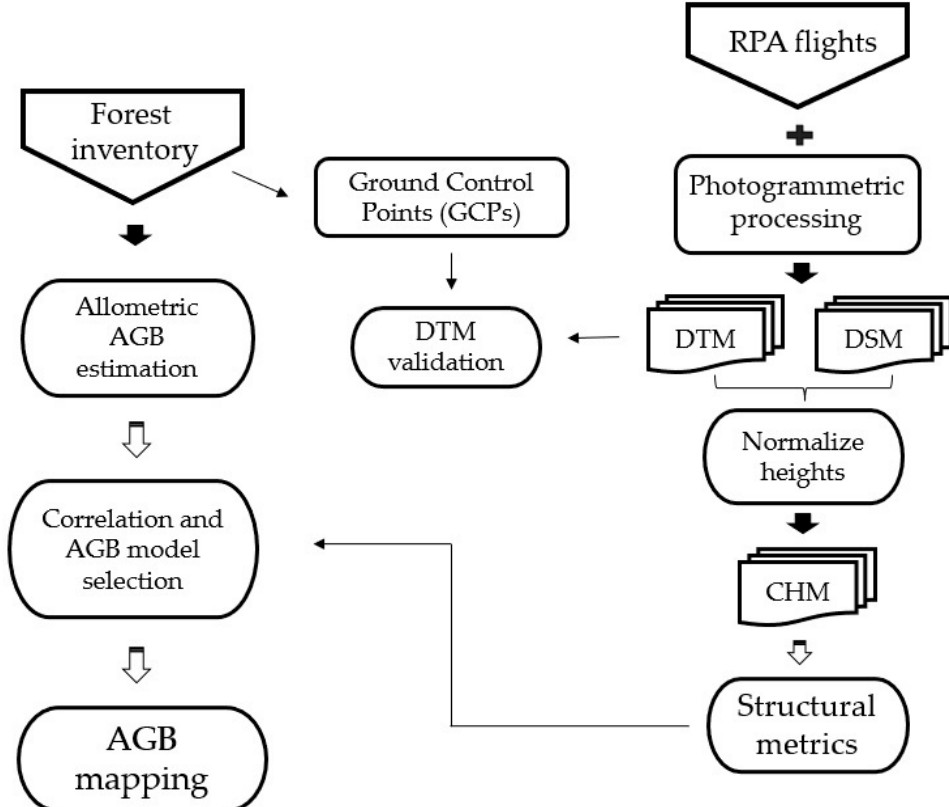

**Figure 3.** Methodology of the workflow used in the current study for modeling and mapping AGB in three areas of cerrado sensu stricto in the Federal District.

Following the assumptions of multiple linear regression, such as normality and homogeneity of variance [42], all variables were subjected to normality (Shapiro–Wilk) and homoscedasticity (Bartlett) tests. Then, the correlation (r) between AGB and metrics was calculated, selecting those that were significant (*p*-value < 0.05) and greater than 0.5 (r > 0.5). Model fitting used the stepwise procedure of selecting the most significant variables for

regression, employing the variance inflation factor (VIF), with an upper limit of 5, to remove collinear predictor variables.

The following precision measures were used for the selection of the best-fitted equation: adjusted coefficient of determination ($R^2$-adj); absolute and relative root mean squared error (RMSE and RMSE%, respectively); graphical distribution of the residuals; graphical distribution of observed versus predicted values. Model validation was performed using the k-fold cross-validation method [43] with 5 groups, including an analysis of $R^2$ values and root mean squared error (RMSE). This cross-validation tool was used to model and validate the regression model, maximizing the available data. For regression fitting, selection, and validation, R software [44] was used.

The selected model was applied over the areas of cerrado sensu stricto covered by the RPA flights, in order to map the biomass stock of the shrub stratum. The selected photogrammetric metrics were spatialized to a grid with 30 m × 30 m cells for the application of the mathematical model [26], using ArcMap software [40].

## 3. Results

### 3.1. Digital Terrain Model (DTM) Validation

The altimetric values captured at the 131 ground control points (GCP) using TechGeo Zenite GPS (II) were highly correlated with the altimetric values estimated by the digital terrain model (DTM), which was generated via photogrammetry of the digital images captured with the RPA (r = 0.99, *p* < 0.01, RMSE = 4.6 m). Assuming an average elevation of 1070 m for the study areas, the associated error represents an acceptable margin of error of 0.4% (Figure 3). The high correlation seen between the data demonstrated that a DTM generated using photogrammetry is able to describe the altimetric variations of the cerrado sensu stricto areas accurately. The clusters of elevation were produced by the elevation differences between the three study areas (Figure 4).

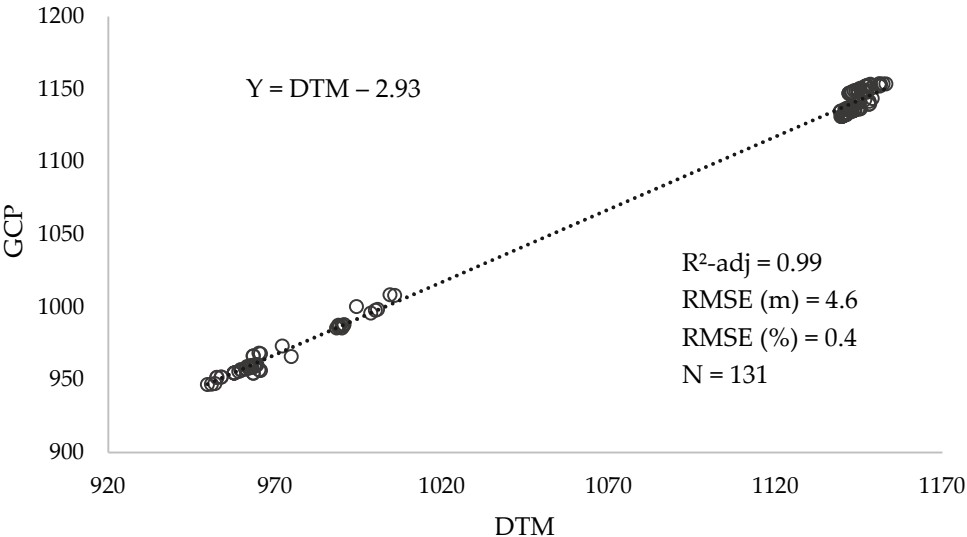

**Figure 4.** Correlation of the altimetric values, captured at ground control points (GCP) with TechGeo Zenite GPS (II), and with the digital terrain model (DTM) generated via the photogrammetry of digital images captured with RPA, at three cerrado sensu stricto sites in the Federal District.

### 3.2. Forest Inventories

The inventory of total estimated AGB was 17.83 Mg ha$^{-1}$ ($\pm$3.33 Mg ha$^{-1}$) at FAL, with 2.51 Mg ha$^{-1}$ ($\pm$2.82 Mg ha$^{-1}$) and 36.93 Mg ha$^{-1}$ ($\pm$7.19 Mg ha$^{-1}$) at the anthropized and preserved sites of ESECAE, respectively. The average biomass value of the plots established in the field at the three sampled sites was 18.3 Mg ha$^{-1}$ ($\pm$13.3 Mg ha$^{-1}$), ranging from 0.1 Mg ha$^{-1}$ (minimum) to 47.6 Mg ha$^{-1}$ (maximum) (Table 3). The wide range of variation of the AGB values and the high coefficient of variation (CV%) reflect

the heterogeneity of the vegetation in the sampled areas, which are currently in different succession phases. However, this heterogeneity of the woody vegetation of the cerrado sensu stricto is a common characteristic of the phytophysiognomy of the environment, in which the structure can vary from sparse cerrado sensu stricto to densely vegetated cerrado sensu stricto [5].

**Table 3.** Descriptive statistics of the AGB and density of woody vegetation (shrub and tree) sampled during forest inventory in three areas of cerrado sensu stricto in the Federal District (*n* = 37).

| Descriptive Statistics | Observed AGB (Mg ha$^{-1}$) | Density (N ha$^{-1}$) |
|:---:|:---:|:---:|
| Minimum | 0.10 | 20 |
| Maximum | 47.6 | 3020 |
| Mean | 18.3 | 1502.4 |
| CV (%) | 72.9 | 60.1 |

In ESECAE's anthropized cerrado sensu stricto, four outlier plots showed very low biomass values that were discrepant from the rest of the database and were not included in the AGB model. The lowest values of AGB and tree density were in the anthropized cerrado sensu stricto area in ESECAE, which is undergoing a slow process of natural vegetation recovery, due to its past intensive use for agriculture and cattle ranching [33]. In FAL, after 27 years, the different silvicultural treatments applied to the plots have produced heterogeneous structures [34], reflected in the high density of individuals and intermediate values of AGB (Figure 5). The fully protected cerrado sensu stricto site at ESECAE showed a lower density of individuals, when compared to FAL, but also exhibited the highest AGB values (Figure 5).

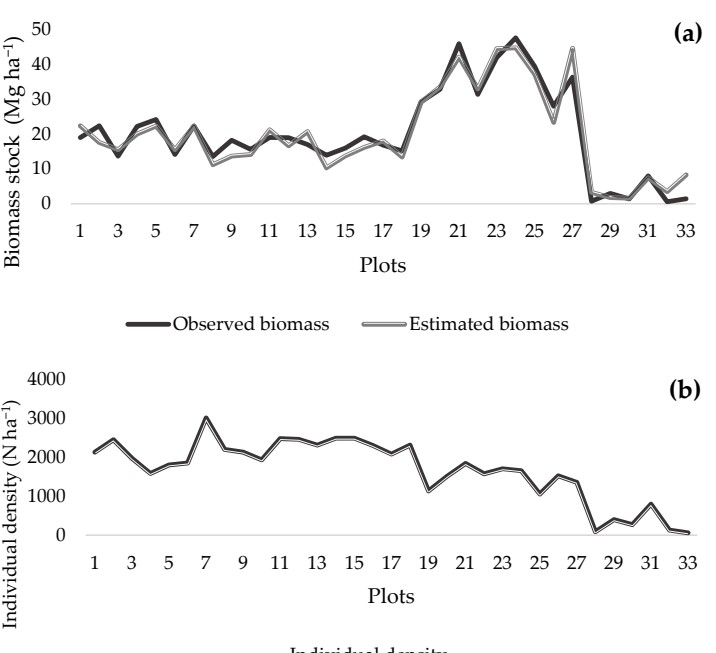

**Figure 5.** (**a**) Observed AGB and estimated AGB values using the photogrammetric model; (**b**) individual density, as observed in the plots in the three areas of cerrado sensu stricto in the Federal District.

### 3.3. AGB Model Selection

Based on the correlation analysis between AGB and the metrics extracted from the digital images captured with RPA, metrics with a correlation lower than 0.5 were excluded. Of the remaining metrics, values were selected to create the AGB model, using HP10, which refers to height at the 10th percentile position, and HP95, which refers to height at the 95th

percentile position. For both variables, the VIF value was equal to 2.9, which is below the established limit of 5, indicating absent or weak collinearity.

The adjusted model with two variables explained 93% of the data variability and presented an estimated error equal to 0.33 Mg ha$^{-1}$ (16%) (Figure 6). The model was validated with 5 groups using the k-fold method, resulting in an R$^2$ value equal to 94% and an RMSE equal to 0.33 Mg ha$^{-1}$ (16%). The residuals showed normal distribution (*p*-value = 0.11, Shapiro–Wilk test) (Figure 7). In both Figures 6 and 7, it is possible to see the grouping of the three sampled sites and the progression of the AGB values in each site.

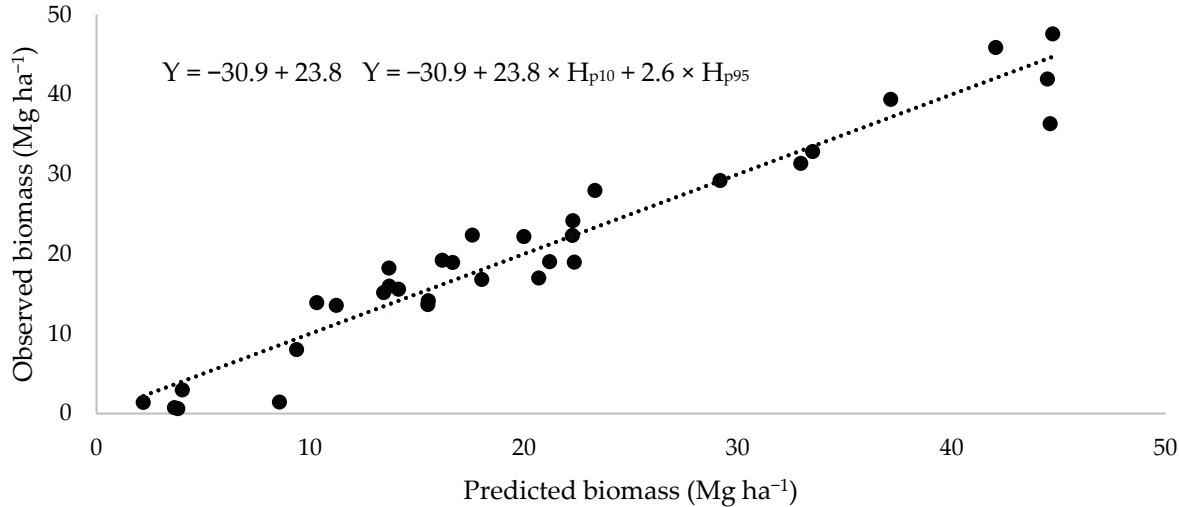

**Figure 6.** Predicted versus observed AGB values in the studied areas of cerrado sensu stricto in the Federal District. Y represents AGB (Mg ha$^{-1}$); H$_{p10}$ and H$_{p95}$ are metrics taken using photogrammetric images, referring to height at the 10th percentile position and to height at the 95th percentile position, respectively.

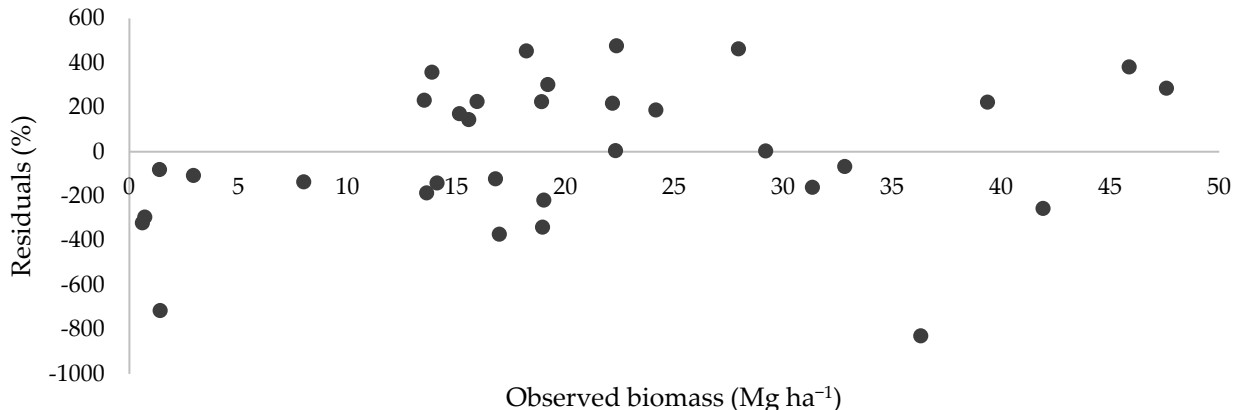

**Figure 7.** Observed AGB versus percentage residuals, obtained via the AGB model (Mg ha$^{-1}$) using photogrammetric metrics, for the tree and shrub stratum of three sites of cerrado sensu stricto in the Federal District.

The maps of AGB distribution generated from the spatialization of the photogrammetric AGB model are presented in Figure 8. This procedure optimized the visualization and understanding of the spatial distribution of forest biomass stocks in tree and shrub stratum vegetation in areas of cerrado sensu stricto, recorded at different stages of succession.

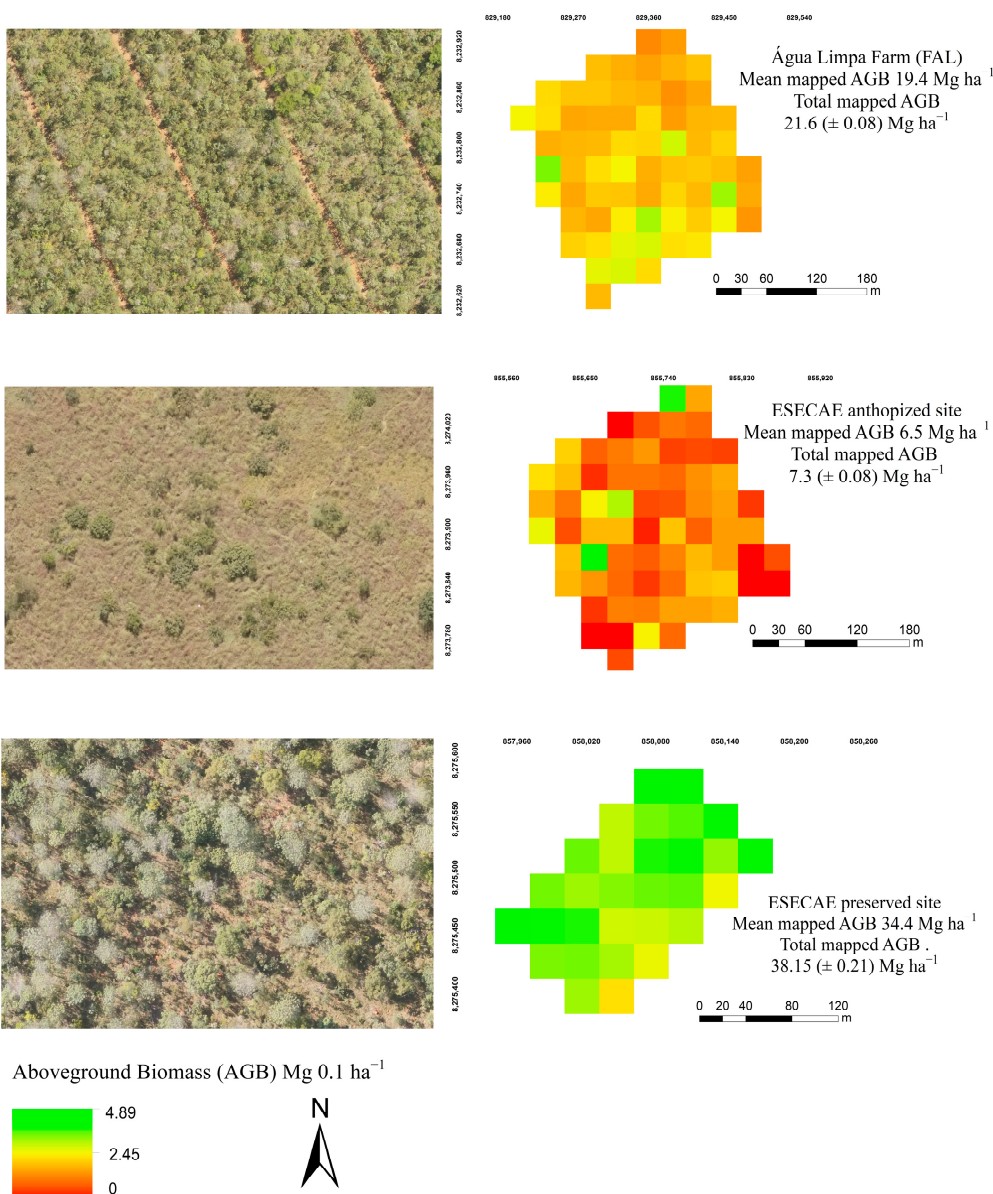

**Figure 8.** Aerophotogrammetrics, AGB distribution maps, and mapped values for the mean and total AGB of the three sites of cerrado sensu stricto, at different stages of succession, in the Federal District: secondary cerrado sensu stricto at the Água Limpa experimental farm (FAL); anthropized cerrado sensu stricto at Águas Emendadas Ecological Station (ESECAE); preserved cerrado sensu stricto at ESECAE.

## 4. Discussion

Radar and laser (airborne or ground-based) sensors can penetrate tree canopies in densely forested areas and can capture centimeter-level information regarding vertical forest structure [15,38]. Light detection and ranging (LiDAR) equipment has been widely used in forest studies, generating 3D point clouds with centimeter accuracy at the stand and individual levels [16], and has been successfully used for estimating biomass stock [45], canopy loss [23], and forest structure [46]. However, despite the high accuracy generated with LiDAR, its acquisition is expensive, especially for organizations in remote regions and developing countries [16], preventing its application on a large scale or for local ecosystem management and monitoring [47]. Furthermore, LiDAR, when coupled to RPA, has flight height limits due to the weight of the unit and low battery autonomy [23].



The inability of passive sensors to penetrate dense vegetation canopies and detect the ground surface is considered a major limitation in their use for AGB estimation in tropical forests [22]. In areas where the vegetation is dense and homogeneous, the lack of light penetrability prevents the capture of forest understory images via passive sensors. Heterogeneous forest landscapes, on the other hand, which exhibit vegetation of different heights and canopy openness, allow the passive sensor to capture images down to ground level [48]. Different studies carried out in tropical forests with very dense vegetation have used digital terrain models of higher accuracy that are generated by active sensors, such as airborne LiDAR or SRTM [15,22,23].

Nevertheless, other studies have already demonstrated the ability of passive sensors to identify the ground surface at sites with sparse and open canopies [22] and with large homogeneous clearings, such as in managed forests [15]. Mlambo et al. [16] demonstrated that the SfM algorithm, when applied in photogrammetry, can generate accurate DTM in areas where the canopy cover does not exceed 50%, or during times of leaf loss in deciduous and semi-deciduous forests, a situation similar to that of the Cerrado biome. This application is possible because the ground surface can be captured in aerial images through these discontinuous canopies, generating consistent digital models [22].

However, even in areas with vegetation suitable for the use of passive sensors and enabling correct reproduction of the ground surface, the topographic complexity must be considered. Areas with steep reliefs influence the accuracy of the DTM, whether it is generated with passive sensors or is used with airborne and ground laser scanning [22]. In addition, there are technical difficulties in performing the flights and scans in these areas. To this extent, the errors in altimetric variation that are associated with DTM are expected in mountainous areas or areas with great relief variations [18].

The metrics generated by the photogrammetric point cloud constitute a valuable information resource regarding the vertical structure of the forest community [47]. In our study, we used a bivariate model in which the selected metrics are percentiles, one that reflects the position of points relative to the normal distribution of the total set of points captured above ground. Given that a single height value does not fully describe any forest stand or community, the percentile represents an index that computes different height values [49] and is, therefore, widely used to describe vegetation structure and model aboveground stocks in different phytophysiognomies [18,23,26,46,49,50].

Therefore, the selection of H_P95, which is representative of the largest trees in the community, is consistent with the ability of the passive sensor to capture surface points that describe the dominant stratum [16,22]. The metric H_P10 complements the model, representing the lowest strata, comprising small trees and shrub vegetation, which is strongly present in the Cerrado savanna physiognomies [5]. Furthermore, the capture and description of the lower strata, by means of the passive sensor, were only possible due to the openness and discontinuity of the canopy. Thus, we understand that the composition of the selected model reflects the savanna physiognomies of the Cerrado, in which scattered trees and shrubs coexist over a graminoid layer [4].

Using a ground laser in the Cerrado biome, Zimbres et al. [50] concluded that metrics based on percentiles clearly described structural differences between the forest (i.e., cerradão) and savanna (cerrado sensu stricto) formations. Similarly, in the same study, the authors adjusted a dry season biomass estimation model for the cerrado sensu stricto based on the 99th percentile, a metric indicative of the highest stratum, along with the 20th percentile, a metric related to vegetation structure at a lower height. Therefore, the most representative strata of AGB in cerrado sensu stricto can be described by high percentile metrics, even when using different remote sensing technologies.

On the other hand, in the eastern Amazon, when using aerial laser (LiDAR) DTM and photogrammetry metrics, d'Oliveira et al. [23] selected the 95th percentile (p95) as the only variable to best predict AGB, representing the dominant stratum in a tropical forest. In forests with dense and continuous canopies, the understory and lower strata are hidden, and cannot be described using passive sensor point clouds [47]. Comparably, these lower

strata do not reflect the aboveground biomass stock found in tropical forests; therefore, a single metric representing the dominant strata (p95) was enough to estimate forest AGB in the study by d'Oliveira et al. [23]. Therefore, the choice of photogrammetric metrics is directly linked to the characteristics of the vegetation studied [48].

The model's uncertainty (>15%) reflects the extent of the database (with 3.7 ha sampled) and the wide range of AGB variability in the sampled population (CV > 70%) [8]. Furthermore, high RMSEs are common in estimates of native forest communities, due to the high natural variability in terms of community structure and composition [10]. Considering that the Cerrado is composed of highly heterogeneous vegetation, even within the same physiognomy [10], and that the average sampling precision of the forest inventories was 38%, the model can be considered accurate.

The accuracy of our predictive model ($R^2$-adj: 93%, RMSE: 16%) was similar to that of other studies using LiDAR in the Cerrado. In an area of cerrado sensu stricto, Bispo et al. [25] generated an estimation model for AGB stock, with an $R^2$-adj value of 93% and an RMSE of 13%. Da Costa et al. [26] also applied LiDAR in the three major Cerrado physiognomies, thereby generating a model for AGB estimation with an $R^2$-adj value of 79% and an RMSE of 33%. Photogrammetry also showed similar accuracy to laser and hyperspectral sensors in discriminating and mapping the species present in savannas [51].

Passive sensors on board a low-cost RPA are capable of generating digital models that are sensitive to landscape changes [52], making it possible to collect regular and updated data on the vegetation in the study areas. Thus, the use of RGB cameras on board an RPA and photogrammetry techniques allow the monitoring of temporal dynamics, connectivity, and the fragmentation of ecosystems. Although RPAs do not offer coverage at larger scales, such as the satellite coverage seen on global levels, they offer viable and cost-effective alternatives for local and regional applications, with excellent temporal and spatial resolution [16,45]. In addition, the photogrammetric models could be upscaled to larger areas using a different combination of sensors, similar to that used by Bispo et al. [25].

AGB is an indirect variable by which to ascertain carbon stocks, which are usually estimated as 50% of dry biomass [53]. AGB models obtained through the use of alternative and cost-effective technologies are crucial and strategic for estimating and mapping carbon stocks [8,54]. The accurate estimation and mapping of the updated distribution of biomass stocks and, consequently, of carbon stocks allow areas undergoing recovery to be monitored, thereby supporting sustainable forest management plans and international negotiations within the carbon market [8,11].

## 5. Conclusions

The use of digital images that are captured with RGB cameras on board an RPA and processed with photogrammetric techniques to generate accurate digital terrain models allowed us to obtain consistent metrics of vegetation height. This made it possible to develop an AGB model that was accurate enough to be applied in different cerrado sensu stricto sites and at different stages of succession.

Our work showed the potential of a low-cost remote sensing platform for accurately estimating and mapping AGB stocks in the Brazilian savanna. Our study represents a step forward in mapping biomass and carbon stocks in tropical savannas, with practical applications for forest managers, technicians, and researchers. Nevertheless, it is essential to evaluate the characteristics of each area of interest, such as canopy cover and altimetric variation, to define the applicability of passive sensors for biomass estimation in other areas of savanna vegetation.

Although our AGB estimation model can be generalized for use in tropical savannas at different successional stages, Cerrado biome woody vegetation is highly heterogeneous. Hence, we suggest that this type of study be implemented in other regions and areas of interest in the Cerrado biome, in order to access information about the different physiognomies and seasonality. This would make it possible to build a more robust biomass stock model that represents the range of variations in the Cerrado.

**Author Contributions:** Conceptualization, M.V.N.d. and A.V.R.; data curation, R.F.P.d.Q.; formal analysis, R.F.P.d.Q., M.V.N.d. and P.A.L.d.A.; investigation, R.F.P.d.Q., M.V.N.d. and P.A.L.d.A.; methodology, R.F.P.d.Q. and M.V.N.d.; project administration, A.V.R.; resources, M.V.N.d. and A.V.R.; supervision, A.V.R.; validation, M.V.N.d.; writing—original draft, R.F.P.d.Q.; writing—review and editing, R.F.P.d.Q., M.V.N.d., A.V.R. and P.A.L.d.A. All authors have read and agreed to the published version of the manuscript.

**Funding:** This study was funded by the Brazilian Coordination for the Improvement of Higher Education Personnel (CAPES, research grant 88882.384103/2019-01) and the University of Brasília (DPG/UNB N°07/2021; PROAP/CAPES/UNB 2022).

**Data Availability Statement:** The data presented in this study are available on request from the corresponding author. The data are not publicly available due to privacy restrictions.

**Acknowledgments:** This study was supported by the Brazilian Agricultural Research Company (EMBRAPA), which provided the RPA.

**Conflicts of Interest:** The authors declare no conflict of interest.

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
