# Peer review of "Estimation of Aboveground Biomass Stock in Tropical Savannas Using Photogrammetric Imaging"

_drones, doi:10.3390/drones7080493_

Round 1
Reviewer 1 Report
The manuscript submitted by de Queiroz et al., titled “Estimation of aboveground biomass stock in tropical savannas using photogrammetric imaging in the journal “Drones” is within the scope of the journal as well as within the scope of the special issue and section.
The manuscript deals with application of Unmanned Aerial Systems or RPAS for photogrammetric surveying of Savanna areas in Brazil and using this data for estimating the above ground biomass. The manuscript looks interesting but suffers from a few issues. Manuscript is well written. Literature review lacks studies published in last two years. Figures need to be improved in quality of presentation (resolution, boundary lines etc.). Following are the comments.
Literature Review can be further strengthened. UAVs are a trending topic and recent publication needs be cited.
Following studies can be referred:
https://www.mdpi.com/2072-4292/15/3/639
https://www.sciencedirect.com/science/article/pii/S2352938523000794
https://www.hindawi.com/journals/ace/2023/3544724/
Concerning UAV application for Savannas for biomass estimation: you can refer these.
da Costa, M. B. T., Silva, C. A., Broadbent, E. N., Leite, R. V., Mohan, M., Liesenberg, V., ... & Klauberg, C. (2021). Beyond trees: Mapping total aboveground biomass density in the Brazilian savanna using high-density UAV-lidar data. Forest Ecology and Management, 491, 119155. https://www.sciencedirect.com/science/article/pii/S0378112721002437
Levick, S. R., Whiteside, T., Loewensteiner, D. A., Rudge, M., & Bartolo, R. (2021). Leveraging TLS as a calibration and validation tool for MLS and ULS mapping of savanna structure and biomass at landscape-scales. Remote Sensing, 13(2), 257. https://www.mdpi.com/2072-4292/13/2/257
In line 57-58, the initial letters of “canopy height model” should be capitalise.
Research questions needs to be identified clearly at the end of the Literature Review and before the objectives
In Figure 1: It needs to be properly edited. There are different font/sizes used in legend. Also improve the quality of resolution. Also mark the name of country and province.
Line 223-224 : “All variables were subjected to normality (Shapiro-Wilk) and homoscedasticity (Bartlett) tests.” : Please explain to the readers why these tests were done. What is the necessity of these tests?
Figure 7: the unit meters should be denoted as ‘m’ following the SI convention and not ‘M’ as done in the manuscript. It should be corrected.
Results and Discussion sections are OK.
Authors can include a figure showing methodology workflow.
Reviewer 2 Report
The manuscript entitled “Estimation of aboveground biomass stock in tropical savannas using photogrammetric imaging” examined the applicability of unmanned aerial vehicle photogrammetry to estimate the aboveground biomass in tropical savannas and produce its distribution map. The authors derived DTM from photogrammetric point clouds and used it to derived CHM of the study areas. The authors also argued the reliability of UAV data to estimate aboveground biomass. Though the manuscript may be interesting, the originality of the manuscript, in my opinion, is low. As I understand, many studies examined the applicability of UAV for the estimation and mapping of forest AGB with the same methodologies used in this study. The manuscript may need some revisions to be published in the journal. I summarized my comments below.
Abstract - the abstract would be better to written in a logical sequence, e.g., very brief introduction, research aim, methods, key results and findings, and conclusion or your interpretation. I would like to recommend to restructure your abstract.
Keywords – please choose the words not included in the title.
Introduction – I feel that most of the introduction section is more like a general information of RS rather than the information supporting your study objectives. Please try to reduce already known and general information of the RS in the introduction (e.g., Line 24 to Line 68).
Materials and methods – under sub-section ‘forest inventory’, would you mention the information related to tree density, tree size, tree height which could influence the estimation of aboveground biomass using UAV data. What are the major species in the study area? How many species were included in the estimation of AGB? What is the height of underground grass in the study area? According to Figure 2, height of the underground grass is high which might affect the estimation of DTM using point cloud derived from UAV data. How did you address this issue?
Results – In Figure 5, the scale on the X-axis and Y-axis should be same.
Discussion – discussion should be based on the results of your study. In the discussion section, you need summarize the main results of the study, compare it with other relevant study, explain and interpret the main results. However, most of your discussion (e.g., Line 343 – 376) didn’t directly related to your study. It is more like a literature review.
Conclusion – it would be better if the authors highlight the main findings and what are the practical application of your findings in the production of high-quality seedlings. You may also add the limitations and future directions of the research. Authors may also need to revise the conclusion section.
Line 11- ‘shrub-woody stratum’ appears for the first time and it didn’t appear again in the whole text. Please clarify the term because it may confuse the content of the study.
Line 216 - fiel-measured AGB >> field-measured AGB
Line 230 - (RMSE) > RMSE? (RMSE%) > RMSE%? It appears for the first time here. You need to mention the long form.
Line 234 - root mean standard error (RMSE)? Root mean square error?
Line 295 - of AGB values in each. Each what?
Figure 5 – the description in X-axis, Predicted biomassa >> Predicted biomass.
Round 2
Reviewer 1 Report
The authors have incorporated the suggestions. Now the quality of manuscript is quite improved and it can be published in this journal.
Author Response
Point 1: The authors have incorporated the suggestions. Now the quality of manuscript is quite improved and it can be published in this journal.
Response 1: Dear reviewer, thank you for taking the time to review our manuscript and help improve its quality.
Reviewer 2 Report
I don't have further comments. However, please check Table 1. You described that species richness (N/ha). Is it the average number of species in each area? I am not sure it is the correct way to describe tree species richness in this way. Please check it again.
Author Response
Point 1: I don't have further comments. However, please check Table 1. You described that species richness (N/ha). Is it the average number of species in each area? I am not sure it is the correct way to describe tree species richness in this way. Please check it again.
Response 1: Dear reviewer, thank you for taking time to review our manuscript. Table 1 was checked, and we changed the way of presenting the information about species richness. Now we present total species richness, i.e., total number of species in each sampled site.